# Fabrication and Characterization of Lignin/Dendrimer Electrospun Blended Fiber Mats

**DOI:** 10.3390/molecules26030518

**Published:** 2021-01-20

**Authors:** Somaye Akbari, Addie Bahi, Ali Farahani, Abbas S. Milani, Frank Ko

**Affiliations:** 1Textile Engineering Department, School of Materials and Advanced Processes Engineering, Amirkabir University of Technology (Tehran Polytechnic), Tehran 15875-4413, Iran; Alifarahani98@aut.ac.ir; 2Department of Materials Engineering, University of British Columbia, Vancouver, BC V6T 1Z4, Canada; addie.bahi@ubc.ca; 3School of Engineering, University of British Columbia, Kelowna, BC V1V 1V7, Canada

**Keywords:** softwood Kraft lignin, polyamidoamine dendritic polymer, blended fiber mats, solution electrospinning, characterization

## Abstract

Blending lignin as the second most abundant polymer in Nature with nanostructured compounds such as dendritic polymers can not only add value to lignin, but also increase its application in various fields. In this study, softwood Kraft lignin/polyamidoamine dendritic polymer (PAMAM) blends were fabricated by the solution electrospinning to produce bead-free nanofiber mats for the first time. The mats were characterized through scanning electron microscopy, Fourier transform infrared (FTIR) spectroscopy, zeta potential, and thermogravimetry analyses. The chemical intermolecular interactions between the lignin functional groups and abundant amino groups in the PAMAM were verified by FTIR and viscosity measurements. These interactions proved to enhance the mechanical and thermal characteristics of the lignin/PAMAM mats, suggesting their potential applications e.g. in membranes, filtration, controlled release drug delivery, among others.

## 1. Introduction

Lignin is one of the complex natural polymers that mostly exists in the by-products of plants, woods, pulp, and paper [1,2]. It is the second most abundant polymer after cellulose, containing approximately 15% of terrestrial biomass, and is one of the strongest and persistent polymers in Nature [3]. Lignin is a polyphenolic material formed through enzymatic dehydrogenation of three phenylpropanoid monomers [4]. Namely, it contains an aromatic structure with a phenolic ring in its precursor monomers, classified into coniferyl alcohol (G), *p*-coumaryl alcohol (H), and sinapyl alcohol (S) groups [3,5,6,7]. Almost 70% of lignin exists in the thickest layer of cell walls. Of this potion, softwood lignin, existing in gymnosperm plants, commonly consists of 24–33% of the cell wall; whereas hardwoods and angiosperm plants has 19–28% lignin content [1,8]. The various structures of functional groups in lignin are related to the type of plant and extraction methods applied [9]. The organosolv, Kraft and soda processes are among the most common lignin’s extraction methods; more specifically, the sodium hydroxide and sodium sulfide in kraft and soda methods, and organic solvents in the organosolv method, are used for delignification and separation of lignin [10]. The number-average molecular weight (M_n_) of 3000 to 10,000, and weight-average molecular weight (M_w_) of 8000 to 80,000 are common for lignin depending on its type and other effective parameters [11]. Besides various structures and types of lignin, it can reveal some special characteristics, such as antioxidant, antiseptic, bactericide, bacteriostatic, microbial resistance, transportation of material to plant’s tissue and disinfectant agent; whilst it is non-cytotoxic to human cells [12,13]. These characteristics have increased the production and demand for lignin-based products in industries over years. For instance, in 1998, only 1% of lignin was isolated and sold [14]; whilst in 2017, there were almost 70 million tons of wood pulping operation worldwide, of which 30 wt.% of extraction was lignin [15,16]. Among different factors, impurities, unwanted depolymerization reaction and degradation upon heating are among the known challenges that need to be dealt with and optimized during development/processing lignin-based products [1,17].

Environmentally, one of the main advantages of using lignin is its renewable resource of carbon due to phenolic structure [18]. Thus, a wide range of lignin based products in the form of powders and/or fibers are utilized as additives e.g. in. composite and nanocomposite products [1,19]. Lignin, however, individually is not capable to be formed into nanofibers, and it should be blended with hydrophilic and hydrophobic polymers such as poly ethylene oxide (PEO), polyvinyl alcohol (PVA), etc [20]. In addition, the mechanical and thermal properties of composite structures can be improved when lignin is grafted or blended with polymers such as polypropylene (PP) as a cross-linker [21], polybutadiene (PB) and PP as an antioxidant [22], polyaniline as a conductivity modifier [23] and starch materials as a hydrophobic co-plasticizer [24]. Lignin has additionally been used as an anti-UV stabilizer due to its phenolic and other organic structure and three-dimensional network of hydroxyl groups [25]. Next to its advantages, lignin has also disadvantages such as its hydrophobic behavior [26], small particle size, and variation in performance including thermal stability [27], along with content condensation fluctuations during extraction and production [28]. Due to these shortcomings, researchers have aimed to improve lignin characteristics by surface or bulk modification methods [29]. One of the novel/emerging techniques for modification of lignin-based products is blending it with dendritic polymers, as also aimed in this paper.

Dendritic polymers have high potential due to their special design, unique characteristics such as host-guest properties, and nano dimension structures. They are formed from repeating units, named generation numbers, around a central core with multiple branches [30]. The most outstanding characteristic, host-guest properties, comes from the large numbers of functional reactive end-groups as well as hollow interior cavities between the branches. The unique characteristic of dendrimers is their branched topology and multifunctional reactive groups [31]. One of the well-known dendrimers is polyamidoamine (PAMAM) providing amine-terminated functional groups.

Electrospinning is known as an efficient method for fabricating fibers in nano and micro-scales [32]. It is believed that electrospinning of lignin blend with other polymers can provide opportunities to achieve new characteristics for lignin-based applications [33]. For instance, the electrospun lignin can be used for carbon fiber production [20,34,35,36], filtration membranes [37], biorefinery processes [38], biomedical and drug delivery applications [39] and wood panel products [40]. Various parameters, such as surface tension, viscosity, and electrical conductivity can also be tuned through lignin nanofiber production [41]. Due to its chemical structure and properties, lignin has been widely blended for electrospinning with other polymers such as polyethylene oxide (PEO) [42], polyvinyl alcohol (PVA) [43], polyacrylonitrile (PAN) [35], and polyethylene terephthalate (PET) [44]. In essence, the blending helps lignin to become a more spinnable fiber due to its good miscibility.

In order to improve the mechanical properties of electrospun lignin fibers particularly due to swelling in water, the fibers need to be heat-set or thermo-stabilized [45]. In heat-setting process, some properties such as color can change (e.g., lignin mats are normally transformed from a light yellow to dark brown). Chemical reactions such depolymerization and recondensation can also take place as increasing the temperature [46]. Lignin has potential to contribute to condensation reaction with fragments, as well as side chain cleavage of ether linkage [45]. Besides, decreasing phenolic bond and forming new reactions can yield higher molecular weights in heat-setting process. In general, stabilization employ a higher temperature range than the heat-setting process, in order to achieve material samples with higher strength and modulus [46].

The aim of the present paper is to evaluate the feasibility of electrospinning and characterization of a novel blend of PAMAM dendritic structure with softwood Kraft lignin (SKL). The heat-setting and thermo-stabilization processes of the SKL nanofibers are performed and analyzed in the presence of PAMAM dendritic polymer for the first time. The interaction between lignin and the polymer host are analyzed by FTIR, TGA, mechanical properties, surface charge, and viscosity measurements. In addition, the morphology of the samples has been examined using scanning electron microscopy (SEM). Overall, the results to follow confirmed some promising characteristics of the synthetized lignin/PAMAM electrospun fibers, with potential industrial applications in e.g. membranes, filtration, and controlled release drug delivery (under development by the authors).

## 2. Results and Discussion

### 2.1. Morphology Analysis

The SEM images of electrospun lignin, lignin/PAMAM (1wt%) and lignin/PAMAM (2wt%) are shown in Figure 1. As seen from the images, and also noticed during processing, after increasing the concentration of dendrimer from 1% to 2wt%, it was not possible to continue the electrospinning process. This is namely because by increasing the PAMAM concentration, the viscosity of the main solution reduces significantly and the fiber fabrication is halted. Similar behavior was reported for other polymers [47]. Hence the optimum concentration of PAMAM in the present blending was assumed to be 1% and all subsequent experiments on the lignin/PAMAM employed the latter percentage. In the remainder of the text, the abbreviation ‘lignin/PAMAM’ will refer to 1wt% of PAMAM.

The fiber diameters of lignin and lignin/PAMAM were measured before and after thermal-treatment (heat-setting) using SEM, as illustrated in Figure 2. It was noted that the average diameters of the lignin/PAMAM nanofibers in both heat-set and not heat-set samples were less than the pristine lignin nanofibers, due to the reduction of surface tension in electrospinning procedure by cationic charges of PAMAM polymers. PAMAM can act as a poly-electrolyte and reduce the diameter of electrospun fiber [48]. In addition, chemical interaction between cationic functional groups of PAMAM and anionic hydroxyl groups in the lignin could be another reason for decreasing fiber diameter (at about 10%). Moreover, per Table 1, the total average diameters of individual lignin and lignin/PAMAM nanofibers were seen to be higher under heat-setting when compared to their respective not heat-set samples; mainly due to dehydration and possible reaction of the hydroxyl group in lignin and lignin/PAMAM. This diameter difference can also be a result of the depolymerization and recondensation processes taking place during heat-setting [46].

Figure 3 reveals the SEM images of lignin and lignin/PAMAM after the (thermo)stabilization process (TSP). The TSP is commonly used to prepare the carbon nanofibers (CNF) [49]. In this process, the higher temperature compared to the heat-setting process causes more diameter reduction (Table 1). The condensation procedure between amino terminated groups of PAMAM (−NH2) and hydroxyl groups lignin (−OH) could not only reduce the fiber diameter, but also make a strong interaction between dendritic groups and the lignin structure.

### 2.2. Mechanical Properties

Mechanical properties of electrospun mats with and without PAMAM before and after heat-setting were illustrated in Figure 4 and summarized in Table 2. The results reveal that the tensile stress (σ_y_), strain at break (%), and the elastic modulus (E) of the both heat-set lignin and heat-set lignin/PAMAM nanofibers have been reduced compared to their respective untreated mats.

While past research had revealed a tenacity enhancement of lignin electrospun fiber after heat-setting (due to more cross-linking and chemical interaction between molecular chains in lignin-based polymers [45,46]), other researchers have reported tenacity reduction of the fibers because of becoming more brittle [50]. Hence, it appears that the processing conditions, such as heating temperature, time of heating, and applied pressure have direct effect on the ensuing mechanical and physical properties. However, almost all past investigators have agreed that the heat-setting (compared to untreated state) can improve durability, biological behavior, and the dimensional stability of the fiber mats [51]. Specifically focusing on the lignin electrospun samples, Table 3 clearly reveals a complex dependency between the heating condition and the ensuing mechanical properties. For instance, both tensile stress and strain at break of the lignin/PEO blend have increased after thermo-stabilization and carbonization process [36], while for another lignin based carbon fiber, the tensile stress has increased and the strain at break has decreased after the heating procedure [52]. In another work, the molecular weight of lignin reduced after heat treatment. These results indicate that heat treatment may degrade the macromolecular structure of lignin to a noticeable extent [45]. The cleavage of the propane side chain (Cγ-C_β_-Cα) from the phenol skeleton (C6) in lignin structure has also been reported due to the heat treatment [46].

In the present work, although there was a sharp drop in the stress-strain response in the case of heat-set samples, the effect of presence of PAMAM on both heat-set and untreated mats was clearly significant (when compared to the bare lignin mats). As shown in Table 2, both (σ_y_) and (E) have increased when adding PAMAM under each group. The phenolic compounds and hydroxyl groups in the lignin chemical structure react with functional groups of PAMAM; thus, making the blended mat much stronger (stiffer under tension). In particular, the strong interaction between amino terminated groups of PAMAM (−NH_2_) in dendritic groups and hydroxyl groups in lignin(−OH) would be another main reason for increasing the tensile stress and elastic moduli for lignin/PAMAM compared to bare lignin electrospun fiber mats.

### 2.3. Surface Charge

Zeta potential can provide important information about surface chemistry and surface charge of electrospun mats, which is relevant to various applications such as water treatment filters, pharmaceutical, environmental protection, paint industry, among others [53]. Zeta potential describes the electrical charge and interactions of small particles at the surface which are dispersed in a material and can indicate the physical stability of the system in different states. In addition, zeta potential and subsequent process optimization can assist to reduce surface energy and improve related applications [54,55].

In this work, different synthesized mats were measured for their surface electrical charges. The surface electrical charge of the lignin was completely negative due to abundant hydroxyl and phenolic compounds in its structure as well as hydrogenation and decarbonylation alkyl groups [56]. Instead, the NH_3_^+^ groups in the PAMAM revealed a positive zeta potential. The zeta potential of lignin was found to be in the range of −23.28 mV, while it increased to −20.72 mV after adding 1% PAMAM based on the lignin weight. This increasing trend confirms that only 1wt% of PAMAM leads to reactions between amine groups in PAMAM with some negative hydroxyl groups of lignin, hence a more electrostatic interaction and leading to a higher elastic modulus in the lignin/PAMAM mat. Dallmeyer et. al [36] also revealed that inter-fiber bonding significantly enhanced the mechanical properties of their electrospun fabrics.

### 2.4. Thermogravimetric Analysis (TGA)

The thermogravimetry (TG) or thermogravimetric analysis (TGA) and derivative thermogravimetry (DTG) curves of lignin and lignin/PAMAM mats, before and after heat-setting, are shown in Figure 5a,b, respectively. The main characteristic and important peaks are also illustrated and compared in Table 4. Generally, lignin thermal decomposition is a complex phenomenon and occurs by various competing reactions [57]. The lignin’s main mass loss peak (according to its type and extraction) commonly appears at about 370–395 °C. The first peak appears in the range of 150–270 °C corresponding to the phenolic compounds of lignin [58]. The weight loss before 150 °C is due to the evaporation of water and small amount of dehydration reaction. Before TGA analysis, all samples were pre-treated by heating at 50 °C for 24 h. Thus, it could not see any water evaporation. At 380 °C, there is a peak related to hydrogenation of methoxy groups on the aromatic ring, and consequently the evolution of methanol. Methane is another product that can release at ~350 °C and it ends at 420–430 °C. Furthermore, carbon dioxide (CO_2_) starts releasing at ~200 °C up to 640–690 °C due to the cracking of C–O–C and C=O bonds [59]. Carbon monoxide (CO) releases from 220 °C to 790 °C due to decarbonylation reaction of alkyl side-chain carbonyl terminate andthe reaction of formaldehyde with free-radical coupling reactors in lignin [60,61].

The results revealed that the synthesized lignin fiber mat with PAMAM has degraded at slightly lower temperature than the bare lignin fiber mats, while the weight of residue left by the lignin/PAMAM mats was higher than that of bare lignin mat, in both untreated and heat-set conditions due to the reaction between lignin and PAMAM dendrimer as shown in Table 4 and Figure 5. As almost PAMAM dendrimer decompose at about 450 °C, onset decomposition temperature start at lower temperature [62]. The interaction between amine groups in PAMAM and hydroxyl groups in lignin has caused more resistant to high temperature. This indicates that lignin/PAMAM would be more thermally stable than the pristine lignin. Similar trends were also reported for halloysite nanotube/PAMAM dendrimer [63]. In Table 4, the comparison between the first and second notable peaks in the DTG indicates the positive effect of dendrimer along with its rich functional group on the mat thermal stability. This effect confirms that using dendrimer in lignin mats can be a sufficient means for production of carbon fibers from lignin along with desirable thermo-mechanical properties.

### 2.5. Fourier Transform Infrared Spectroscopy (FTIR)

As shown in Figure 6 and Table 5, the bands of lignin contain multiple functional groups. The peak at 1429–1509 cm^−1^ and 1601 cm^−1^ is related to C–C aromatic rings [64]. The lignins’ alcohol peaks showed different bands (at 1260–1270 cm^−1^ and 1330–1375 cm^−1^) related to the guaiacyl and syringyl moieties, respectively. In addition, the carbonyl group is seen at 1700–1715 cm^−1^ [65]. The absorption bands of water appeared in a range of 1300–1800 cm^−1^ and 3500–3964 cm^−1^. The existence of CO is verified by bands at 2112 and 2180 cm^−1^, while CO_2_ appeared at 2217–2391 cm^−1^ [61]. The most notable band in lignin is O–H due to the presence of the alcoholic and phenolic structure of lignin at 1300–1400 cm^−1^. The absorption bands at 2850–3200 cm^−1^ indicate the hydrocarbons, such as methane [58]. Furthermore, the peak of C=C bonds of aromatic rings in lignin appeared at 1595 cm^−1^. On the other hand, PAMAM, which is rich in amine terminated functional group, produced pick bands at 1550–1640 cm^−1^. The band at 3100–3500 cm^−1^ elucidated the amino groups of PAMAM. These absorptions appeared at 1630–1680 cm^−1^ in the FTIR spectroscopy of lignin/PAMAM (Figure 6) providing a proof for the effective reaction between lignin and PAMAM. Also shown in Figure 6b,c), a new peak appearing at 1655 and 1705 cm^−1^ could further confirm the reaction of amine group of PAMAM and hydroxyl group of lignin [66].

### 2.6. Viscosity

To assess the interaction of lignin with PAMAM, lignin and lignin/PAMAM solutions were prepared and their viscosities were measured immediately and after 5 days (to assess the progress/decay of the chemical interactions e.g., due to storage condition etc), as shown in Figure 7. Furthermore, due to the relationships between viscosity property at the time of processing and ensuing fiber diameter through electrospinning, the SEM images of the electrospun fibers after 5 days were illustrated in Figure 8 (see also Figure 1 and Figure 2 for t = 0). In general, the viscosity would be increased by increasing more entanglement between polymeric chains, while it is reduced due to the less entanglement [47]. The results in Figure 7 reveal that the viscosity for both lignin and lignin/PAMAM have decreased after 5 days (this would recommend that in practice a stored solution should not be used for future electrospinning). It seems that the unique structure of lignin, which is rich in phenolic rings and hydroxyl groups, causes the depolymerization of lignin in DMF as well as the transfer reaction to DMF solvent [67]. On the other hands, the molecular weight distribution of the solution shifts toward higher values over time, as the smaller species are “consumed” to form larger molecules [68]. The viscosity measurement results in Figure 7 showed an increase of viscosity in the presence of PAMAM, at both initial and 5-day instances. The interaction between PAMAM and lignin has made the solution denser, due to more molecular chain entanglement. By increasing the PAMAM concentration from 1wt.% to 2wt.%, the viscosity of the base solution has increased by 44% and 55%, respectively.

### 2.7. Reaction Mechanism between Lignin and Amine-Terminated Dendrimer

Despite attempts made to define the various structural elements of lignin, the clear chemical structure of softwood Kraft lignin (SKL) is still under investigation due to the complexity of various types [68]. Even though the most quantity of defined structure belongs to the methoxy groups, SKL has various inter-unit bonds, such as arylglycerol-β-aryl ethers, phenyl coumarans, pinoresinols, lignin carbohydrate α-benzyl ethers, stilbenes, aryl enol ethers, secoisolariciresinols, as well as end groups, such as cinnamyl alcohols, arylacetic acids, arylhydroxyacetic acids, aryl ethyl ketones, arylpropanols, arylhydroxyethyl ketones, aromatic aldehydes, cinnamyl aldehydes and functional groups, such as aromatic C–H, methoxy, quinones, and also hydroxyl groups, such as carboxylic OH, aliphatic OH, *o*-disubstituted phenols (including *o*-substituted catechols), *o*-monosubstituted phenols, and phenolic OH [69]. It is well-known that the amination reaction of lignin is regularly done through Mannich reaction with amino group and formaldehyde [70]. In the absent of formaldehyde in lignin/PAMAM solutions, the reaction between PAMAM and main segment of lignin (methoxy groups) is carried out with hydrogen bonding and ionic interaction between hydroxyl group in lignin and amino terminated group in PAMAM. The regular reaction occurs between R–OH and R’-NH_2_ where R and R’ come from the substitutions of lignin and PAMAM, respectively. The effect of these interactions could be realized by presence of new peaks in the FTIR, strength enhancement, and viscosity variation results presented in the previous sections. As it is very complicated to demonstrate the reaction between SKL and PAMAM, instead the chemical interaction between guaiacyl (G) lignin and lower generation, 1st generation, of PAMAM is illustrated in Figure 9.

## 3. Materials and Methods

### 3.1. Materials

Softwood Kraft lignin (SKL) was supplied by the FP Innovations (Vancouver, BC, Canada) and used with no further purification. *N*,*N*-Dimethylformamide (DMF, 99.9%), polyethylene oxide (PEO) (M_w_ = 900 kDalton) was purchased from Aldrich (St. Louis, MO, USA). Polyamidoamine (PAMAM) dendritic polymers was donated by Delta Innovative Company (Dolsk, Poland).

### 3.2. Preparation and Properties of Electrospinning Solutions

Electrospinning solutions were prepared by dissolving an exact amount of SKL into the DMF solvent in the presence of PEO at 1 %wt. based on the lignin powder. For lignin/PAMAM samples, 1% and 2 wt. % of PAMAM based on the lignin powder were dissolved in DMF solvent, while lignin and PEO were separately dissolved into DMF solvent. After 2 h mixing, these two solutions were mixed together and continued for another 1 h before electrospinning. The solution concentration for all samples was 30 wt%. The electrospinning procedure was applied in an electrospinning distance of 15 cm with an applied voltage range of 17 kV and with 0.01 mL/min pump speed.

#### Heat-Setting and Stabilization Procedures

After drying the mats at room temperature for 10 h, for preparing the heat-set samples, the samples were heated in oven (air atmosphere) to 150 °C and kept for 120 min. While, for the stabilization procedure, the samples were heated at 4 °C/min heating rate to 250 °C and kept for 90 min in a gas chromatography oven (Hewlett Packard 5890 Series II, Hewlett-Packard Company, Wilmington, DE, USA).

### 3.3. Characterization of Composite Membranes

#### 3.3.1. Morphology Measurement

The morphology of the electrospun nanofibrous mats was measured by a scanning electron microscope (SEM, S-2300, Hitachi, Schaumburg, IL, USA) with an acceleration voltage of 5 kV. All the samples were gold coated before the SEM observation. The average diameter of nanofibers was determined by analyzing the images via Image J.

#### 3.3.2. Mechanical Properties

The mechanical properties of the electrospun mats were evaluated using a multipurpose micro-tensile tester (KES-G1, Kato-Tech Co. Ltd., Minami-ku, Kyoto, Japan) with a 5 kg capacity force transducer. Strip-specimens (0.5 cm in width × 5 cm in length) were tested with 5 replications for each sample at a cross-head speed of 0.2 cm/s. The strain was calculated by dividing the displacement by the gauge length. Stress was obtained according to:(1)$$\sigma_{\mathrm{specific}}\;=\;\mathrm{Specific}\;\mathrm{Stress}\;(N/\mathrm{Tex})\;=\;\frac{\mathrm{Force}\left(N\right)/\mathrm{Width}\;\mathrm{of}\;\mathrm{specimen}\;\left(\mathrm{mm}\right)}{\mathrm{Areal}\;\mathrm{density}\;\mathrm{of}\;\mathrm{specimen}\;\left(g/m^{2}\right)}$$
1N/tex = 145,000 psi = 1 GPa(2)

Load cell setting was g/10v, deformation rate was set at 0.02 cm/s, the specimen length was 5 cm, and the mass of specimen was calculated separately for each sample. The engineering stress σ_Eng_ was achieved from the equations below (where V represent the signal voltage read through the machine):(3)Deformation(cm)=Deformation rate×∆t
(4)$$\mathrm{Strain}\;\left(\frac{\mathrm{cm}}{\mathrm{cm}}\right)=\frac{\mathrm{defromation}}{\mathrm{specimen}\;\mathrm{length}}$$
(5)Load(g)=Load cell setting ×∆V

Areal density was calculated as:(6)$$\mathrm{Areal}\;\mathrm{Density}\left(\frac{g}{m^{2}}\right)=\frac{\mathrm{Sample}\;\mathrm{Mass}\left(g\right)}{\mathrm{length}\left(m\right)\times \mathrm{width}\left(m\right)}$$
(7)$$\sigma_{\mathrm{specific}}\left(\frac{g}{\mathrm{tex}}\right)=\frac{\mathrm{load}\left(g\right)}{\mathrm{ArealDensity}\left(\frac{g}{m^{2}}\right)\times \mathrm{Width}\left(\mathrm{mm}\right)}$$

Converting the specific stress to engineering stress can be achieved by:(8)$$\sigma_{\mathrm{Eng}}\left(\mathrm{MPa}\right)=\sigma_{\mathrm{specific}}\left(\frac{g}{\mathrm{tex}}\right)\times 9.81\times \rho_{\mathrm{polymer}}\;(g/{\mathrm{cm}}^{3})$$

#### 3.3.3. Surface Charge

A surface zeta potential (SZP) electrode was used for analyzing the charge on the fibrous mats’ surface by a NanoBrook instrument (Brookhaven Instruments Corporation, Holtsville, NY, USA). There was a special holder to attach nanofiber mats in zeta sizer equipment to measure the surface charge. Measurements were accomplished by attaching the sample to a horizontal place situated perpendicularly between two electrodes in special holder. The sample and electrodes were placed into a diluent containing probe particles for which the charge is pre-known.

#### 3.3.4. Thermogravimetric Analysis (TGA)

Thermogravimetry is a known method to assess the thermal resistance of the samples and to obtain useful properties like oxidative degradation, dissolution, decomposition [71]. Thermogravimetric analysis (TGA) was carried out using a thermogravimetric analyzer (Q500, TA Instruments, New Castle, DE, USA). Samples were heated at 20 °C/min from room temperature to 600 °C in a dynamic nitrogen atmosphere (flow rate = 60 mL/min). Before TGA analysis, all samples were pre-treated by heating at 50 °C for 24 h.

#### 3.3.5. Fourier Transform Infrared Spectroscopy (FTIR)

Fourier transform infrared spectroscopy (FTIR) can provide important information on the molecular structure of components of polymeric materials and it is widely used for geological samples [72,73,74]. The mechanism of the FTIR technique is related to transitions between quantized vibrational energy states [72,75]. The FTIR spectra were recorded over the range of 400–4000 cm^−1^ with 64 scans and a resolution of 4 cm^−1^ on a 660-IR instrument (Agilent, Santa Clara, CA, USA).

#### 3.3.6. Viscosity Measurement

The solution viscosity of lignin for electrospinning is deemed an essential factor that need to be controlled during the fabrication process. The concentration of lignin in the solution and time are considered as two main elements affecting the viscosity [52,76]. Here, dimethylformamide (DMF) was chosen as solvent for lignin and lignin/PAMAM experiments. The effect of mixing on the viscosity was assessed after 5 days. The viscosity values were reported by a torsional oscillation-type viscometer (SEKONIK corporation, Tokyo, Japan).

## 4. Conclusions

The blending of softwood Kraft lignin (SKL) and PAMAM dendritic polymer, followed by the solution electrospinning method, was demonstrated in this paper. The optimum percentage of PAMAM in blended electrospun fibers was found to be 1% wt. Heat-setting and stabilization procedure were applied to both lignin and lignin/PAMAM electrospun mats. The chemical interaction between hydroxyl groups in lignin-based mats and amino functional groups in PAMAM dendritic structure were studied by FTIR. The presence of new peaks at 1655 and 1703 cm^−1^ confirmed the presence of new amide bonds. The viscosity of the solution at 1% and 2% wt. PAMAM concentration were measured. The results, both at t = 0, and 5 days after of mixing, revealed increased viscosity by adding the PAMAM concentration; however, electrospinning was feasible using up to 1% wt PAMAM. The higher viscosity of the lignin/PAMAM solution comparing to lignin solution is probably related to the interaction between PAMAM and lignin confirmed by FTIR spectra. Increasing surface charge of lignin/PAMAM was another evidence for the interaction between cationic amino groups in PAMAM and hydroxyl groups in lignin. Furthermore, the mechanical properties of the bare lignin and lignin/ PAMAM mats were analyzed. There were significant enhancements in the tensile stress and elastic modulus of the lignin/PAMAM compared to the lignin mats. The morphology of the electrospun fibers were studied by SEM. A reduction in average diameter for lignin/PAMAM fibers was also realized (compared to pristine lignin), as a result of higher surface tension of the solution in PAMAM/lignin.

The observed enhancements in mechanical and chemical properties of lignin/dendritic polymer fibers appear to be an encouraging step forward for development of low-cost precursors for multifunctional lignin-based fiber products (e.g. for use in membranes, filtration, antiviral activity, controlled release drug delivery, etc.). A more in-depth micro-analysis of the formation of molecular interactions between lignin and dendritic polymers (here PAMAM) will be highly beneficial toward optimizing such applications.

## Figures and Tables

**Figure 1 molecules-26-00518-f001:**
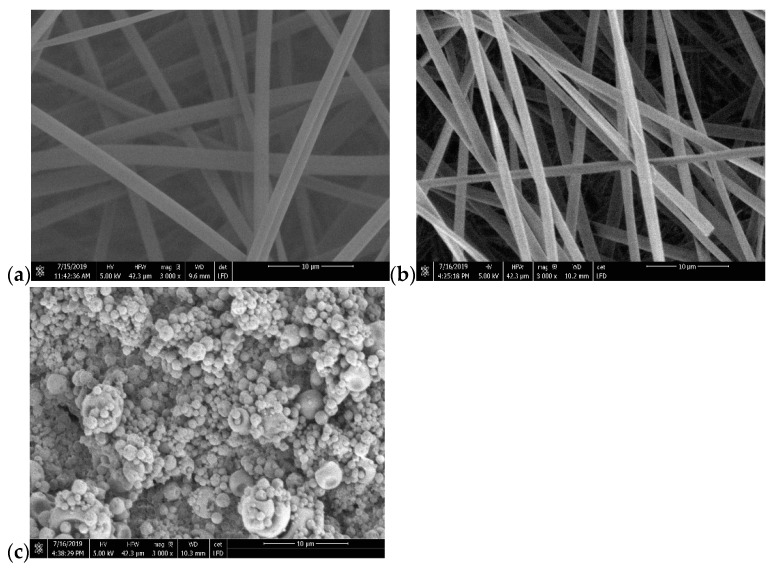
SEM of (**a**) lignin; (**b**) lignin/PAMAM 1wt%; (**c**) lignin/PAMAM 2wt%.

**Figure 2 molecules-26-00518-f002:**
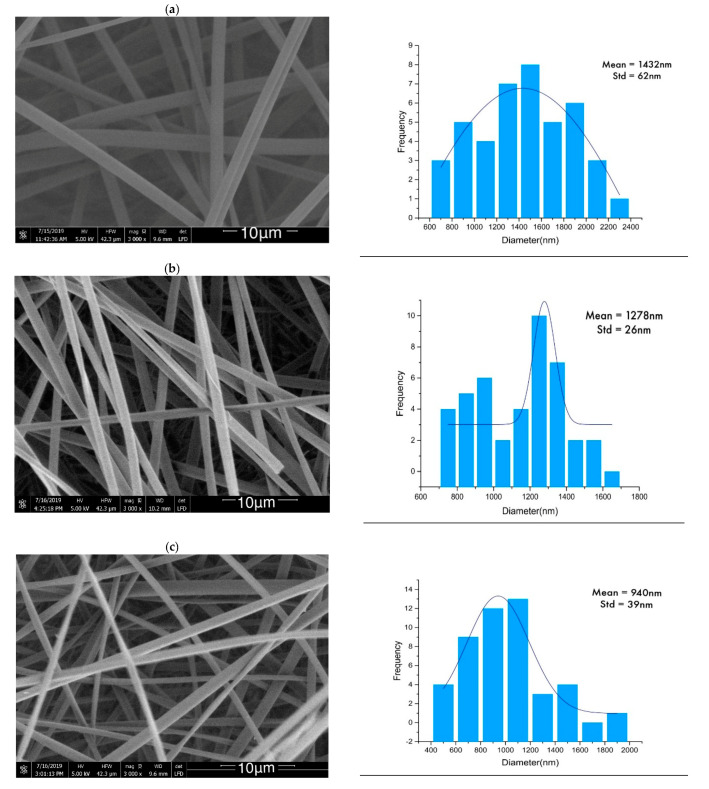
SEM of (**a**) lignin, (**b**) lignin/PAMAM, (**c**) heat-set lignin, and (**d**) heat-set lignin/PAMAM, along with the corresponding fiber dimeter distributions.

**Figure 3 molecules-26-00518-f003:**
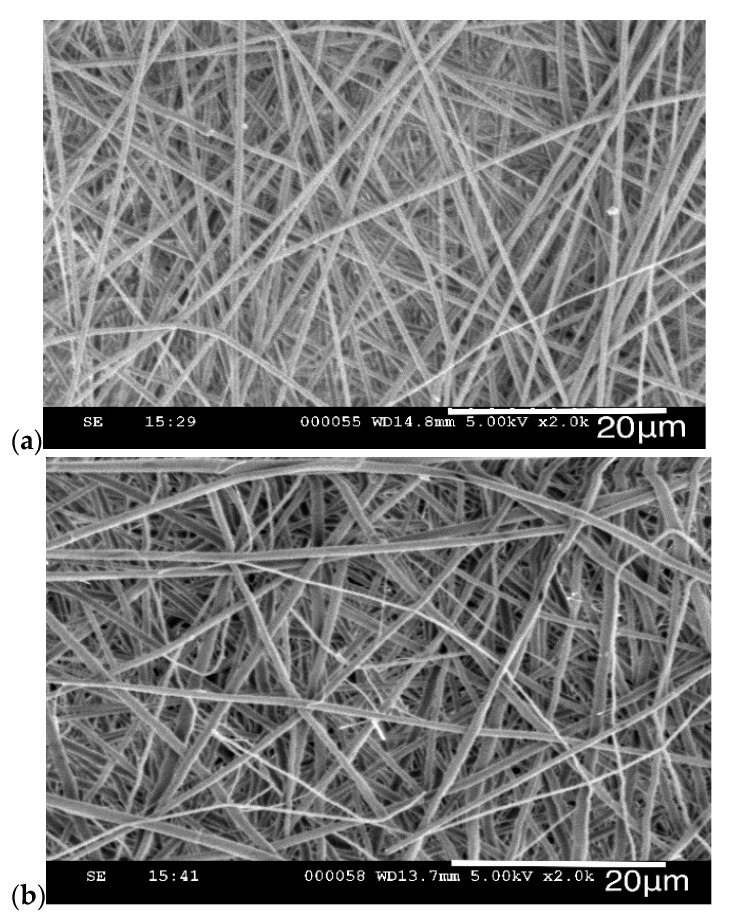
SEM of (**a**) lignin Stabilized and (**b**) lignin/PAMAM stabilized.

**Figure 4 molecules-26-00518-f004:**
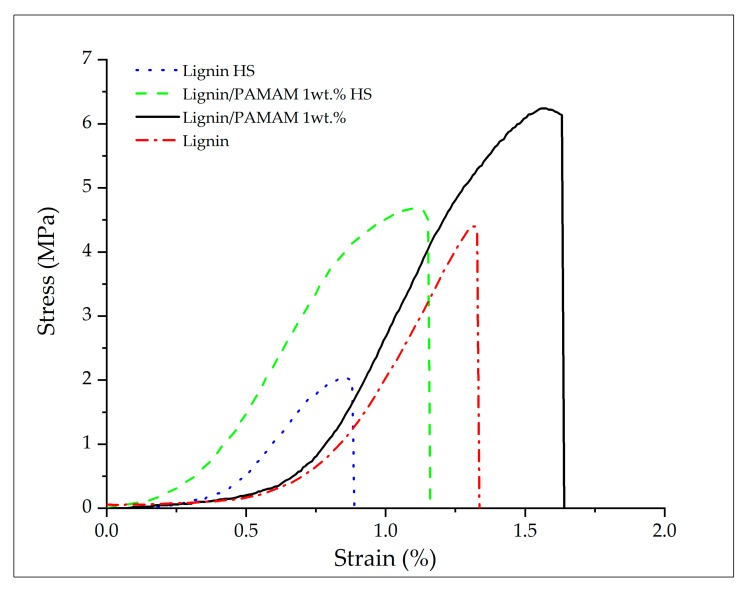
Sample stress-strain response of lignin and lignin/PAMAM electrospun mates before and after heat-setting procedure. Very slight fiber slippage was observed in the beginning of deformation. The vertical lines indicate the breakage onset (material failure) in each of the tests.

**Figure 5 molecules-26-00518-f005:**
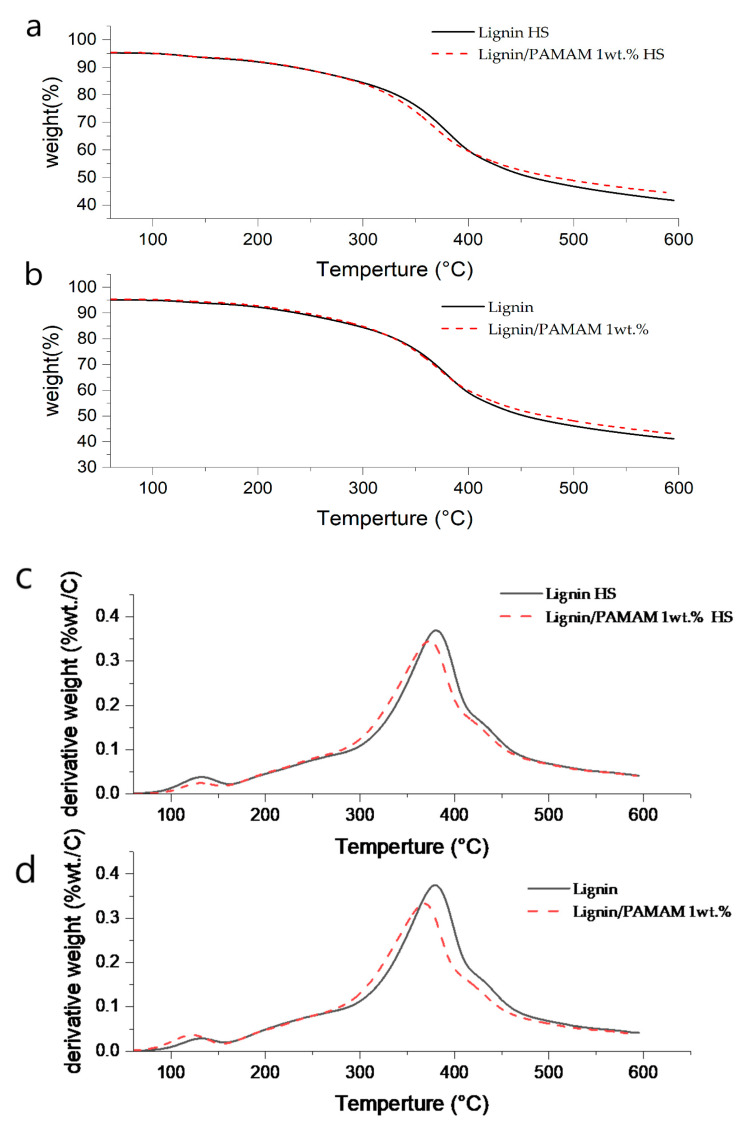
The TG of (**a**) Heatset (HS) samples, (**b**) untreated samples, and DTG of (**c**) Heatset (HS) samples, (**d**) untreated samples curves of the lignin and lignin/PAMAM dendrimer, under untreated and heat-set (HS) states.

**Figure 6 molecules-26-00518-f006:**
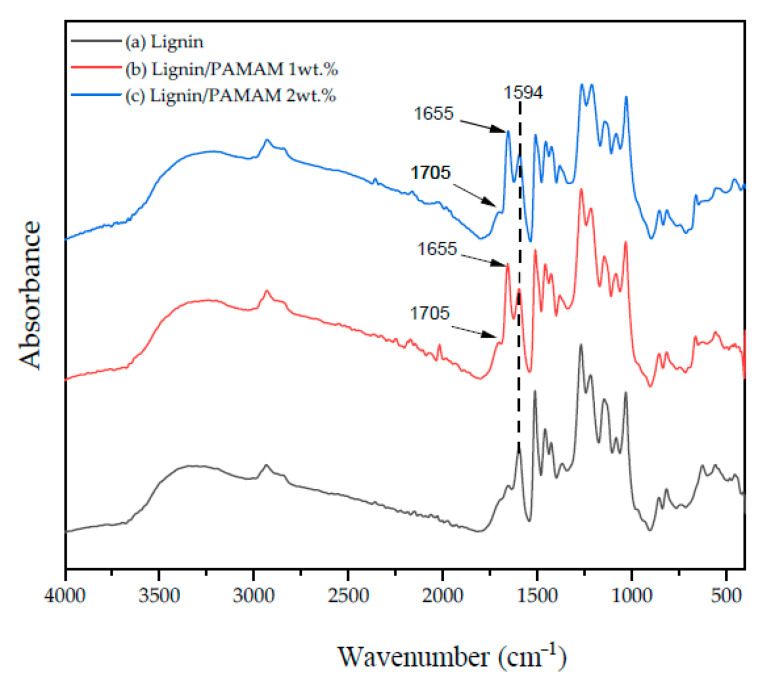
FTIR spectra of (**a**) lignin, (**b**) lignin/PAMAM 1wt%, and (**c**) lignin/PAMAM 2wt%.

**Figure 7 molecules-26-00518-f007:**
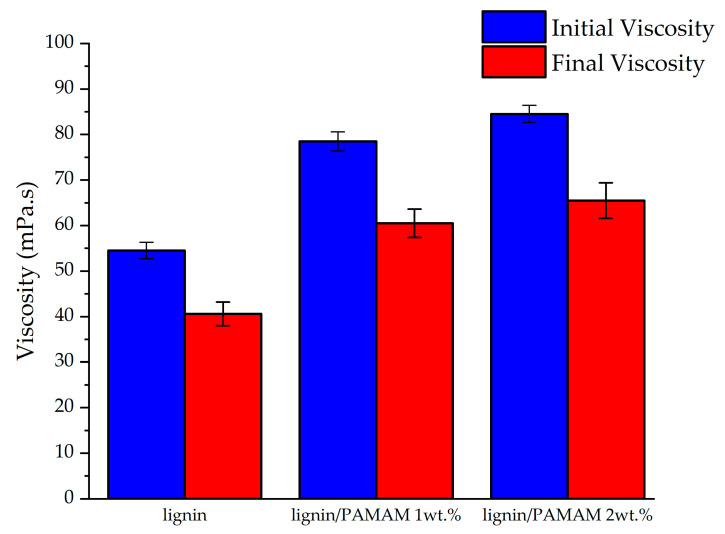
The viscosity of lignin and lignin/PAMAM solutions; initial and final viscosities are related to t = 0 and after 5 days solution, respectively.

**Figure 8 molecules-26-00518-f008:**
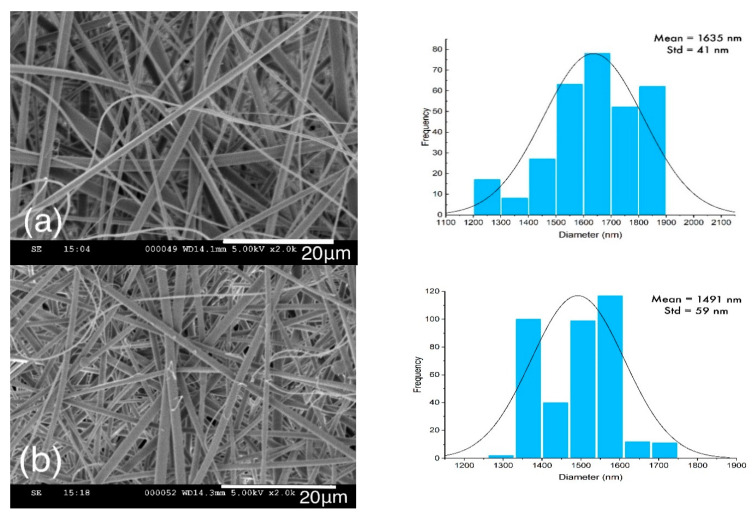
SEM image and diameter distribution of (**a**) lignin and (**b**) lignin/PAMAM 1wt.%, after 5 days.

**Figure 9 molecules-26-00518-f009:**
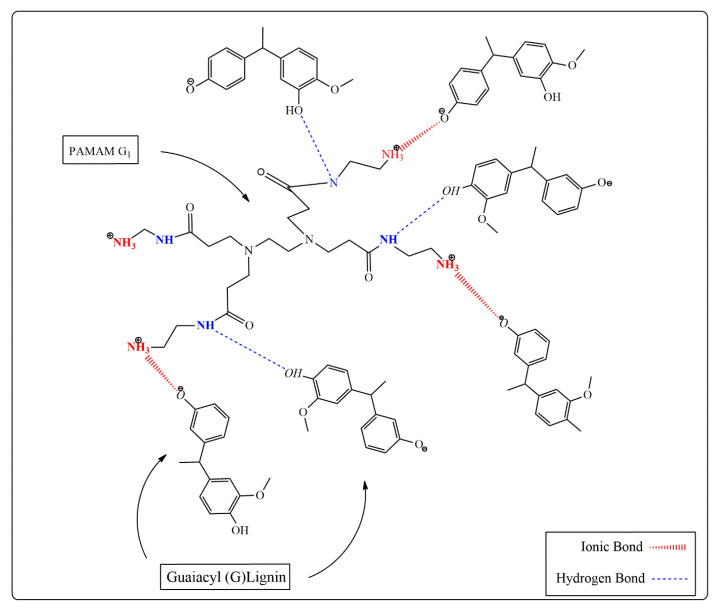
The possible ionic and hydrogen bonding between lignin and PAMAM.

**Table 1 molecules-26-00518-t001:** The average fiber diameter of electrospun lignin and lignin/PAMAM at the states of untreated (pristine), heat-set, and thermo-stabilized.

Sample	Untreated (nm)	Heat-Set (nm)	Stabilized (nm)
Lignin	1432 ± 62	940 ± 39	622 ± 21
Lignin/PAMAM	1278 ± 26	909 ± 41	866 ± 29

**Table 2 molecules-26-00518-t002:** Mechanical properties of the untreated and heat-set lignin nanofiber mats.

Properties	Untreated Lignin Nanofiber Mats		Heat-Set Lignin Nanofiber Mats	
	Lignin	Lignin/PAMAM	Lignin	Lignin/PAMAM
Tensile Stress (MPa)	4.52 ± 0.97	6.30 ± 1.70	2.23 ± 0.41	4.47 ± 0.56
Strain at break (%)	1.32 ± 0.12	1.57 ± 0.27	0.85 ± 0.08	1.10 ± 0.19
Elastic modulus (E) (MPa)	342.42 ± 8.01	401.27 ± 1.73	262.35 ± 6.68	442.57 ± 4.74

**Table 3 molecules-26-00518-t003:** Comparision of the tensile stress, strain at break, and elastic modulus properties of different blended lignin electrospun products from past investigations, depending on the heat-set conditions.

Sample	Heat-Set (HS) Condition	Tensile Stress (MPa)		Strain at Break (%)		Elastic Modulus (GPa)		Ref.
		Before HS	After HS	Before HS	After HS	Before HS	After HS	
Lignin/PEO	Carbonization	32.01 ± 9	74.1 ± 14.6	0.9 ± 0.3	3.01 ± 1.1	4.8 ± 0.6	4.1 ± 1.4	[36]
Lignin/PEO	Thermo-stabilization	17.2 ± 3.4	31.4 ± 6.4	2.3 ± 0.5	3.9 ± 1.4	0.8 ± 0.06	1.3 ± 0.09	[36]
Lignin Carbon fiber 25 wt%	Carbonization	5.13 ± 0.64	45.3 ± 9.93	1.73 ± 0.45	0.7 ± 0.05	0.5 ± 0.07	6.23 ± 1.01	[52]
Lignin Carbon fiber 27 wt%	Carbonization	8.35 ± 1.15	53.1 ± 22	1.5 ± 0.2	0.5 ± 0.07	0.7 ± 0.05	6.8 ± 1.01	[52]
Lignin carbon fiber 25 wt%	Stabilization	8.35 ± 1.15	23.86 ± 4.55	1.73 ± 0.45	3.12 ± 0.61	0.5 ± 0.13	0.9 ± 0.13	[52]
Lignin carbon fiber 25 wt%	Stabilization	8.35 ± 1.15	25.5 ± 4.55	1.5 ± 0.3	1.75 ± 0.3	0.7 ± 0.05	1.9 ± 0.22	[52]

**Table 4 molecules-26-00518-t004:** The characteristic points of TG and DTG curve at two main peaks for different mat groups.

Component	T_onset_ ^a^ (°C)	T_d_ ^b^ (°C)	Residue Weight at T_d_ %	T_f_ ^c^ (°C)	Residue Weight at T_f_ %	α_1_ ^d^	α_2_ ^e^
Lignin	47	380	65.81	596	41.12	0.374	0.029
Lignin/PAMAM	50	367	69.66	594	43.2	0.345	0.038
Lignin (HS)	52	381	66.31	596	41.69	0.369	0.024
Lignin/PAMAM (HS)	51	373	66.38	589	45.54	0.332	0.036

^a^ The decomposition onset temperature, estimated as the temperature for 5% weight loss; ^b^ The decomposition temperature; ^c^ The temperature at final residue weight; ^d^ The derivative weight (%wt./C) at 66% weight; ^e^ The derivative weight (%wt./C) at 94% weight.

**Table 5 molecules-26-00518-t005:** Main chemical elements in lignin samples and their range of evolution, detected by FTIR.

Wavenumber (cm^−1^)	Functional Group	Vibration	Compounds	Refs.
645–690	C=O	Dactyl-zone		[60,65]
974–1058	C–O	Stretching	R–OH	[60,64]
1000–1300	C–O	Stretching		[58,65]
1093–1188	C–C	Skeleton		[60]
1310–1365	C–CH_3_	Bending	Alkyls	[60,65]
1300–1400	O–H	Bending		[61,63]
1513	C=C–OH	Stretching		[65]
1613	C=C	Stretching	Aromatics	[65]
1645–1750	O–H	Bending	H_2_O	[60]
2020–2220	C–O	Stretching	CO	[60]
2210–2390	C=O	Stretching	CO_2_	[60,65]
2750–2990	C–H_2_	Asymmetric Stretching		[58,60]
2904–2979	C–H	Stretching		[64]
2990–3010	C–H	Stretching		[58,64]
3020–3190	C–H	Stretching	CH_4_	[61,64]
3500–3600	O–H	Stretching		[58,64]
3823–3870	O–H	Stretching		[60,64]

## Data Availability

All obtained data has been included in the body of the manuscript.

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
