# Peer review of "Fabrication and Characterization of Lignin/Dendrimer Electrospun Blended Fiber Mats"

_molecules, 2021, doi:10.3390/molecules26030518_

Round 1

Reviewer 1 Report

I continue to judge the manuscript of insufficient quality, both in terms of language (and proof-reading), scientific substance, and clarity.

How does "solution density" depend on chain entanglement (s. conclusions)?

How were "surface charge" and "surface density" measured, and what did these measurements mean?

How does "lignin depolymerization in DMF" explain a reduction in viscosity with time while molecular weights increase with time (pg. 9)?

How are "small molecules consumed with time"? Many open questions could easily be answered by targeted lignin modification (and characterization), and by adding some information on experimental variables. By the way,  hardwood (angiosperm) lignin is not 95% guaiacyl-containing. Amine functionality interacts preferentially with carboxyl groups rather than general (aliphatic) OH groups.

Reviewer 2 Report

Fabrication and Characterization of Lignin/Dendrimer Electrospun Blended Fiber Mats by Somaye Akbari, Addie Bahi, Ali Farahani, Abbas S. Milani and Frank Ko

The authors have demonstrated preparation of a novel blend of PAMAM dendritic structure with softwood kraft lignin by electrospinning. The manuscript needs major revisions before it is considered for publications. 

Major points

“Materials an methods” and “Results and Discussion” are given in not classical order. Step of water evaporation is not visible on TG curves, please provide the beginning of curves in more detailed scale.

Td temperature given in Table 4 has not any physical or chemical meaning.

Deference in residue weight after thermal decomposition does not provide information about thermal stability. The onset decomposition temperature is used for this purpose.

How CO2 was produced under inert atmosphere? The mechanical results (E about 80 GPa) are extremely high for polymer materials.

Please provide mechanical curves for proving.

Why concentrations 1 and 2% were chosen for preparation of composites.

Minor points

The acceleration voltage in SEM experiment was only 5kV, like it is seen in Figures. However, the range 5-20 kV is given. Fig. 5. Y-axis without units. Please check manuscript for typos.

Author Response

Reviewer 2

 “Materials and methods” and “Results and Discussion” are given in not classical order. Step of water evaporation is not visible on TG curves, please provide the beginning of curves in more detailed scale.

>Thank you for your comments. Indeed, we agree with the reviewer comment regarding the “Materials and methods” and “Results and Discussion” classical order. However, we obeyed the template given by the journal. In Molecules journal, first we need to report “Results and Discussion” section, then the “Materials and methods”.

Before TGA analysis, all samples were pre-treated by heating at 50ºC for 24 h. Hence, we did not see any water evaporation. This has been now clarified in the revised text.

Td temperature given in Table 4 has not any physical or chemical meaning.

>Thank you for your comment. Indeed, numerous investigations have explained thermal characteristic by Td. Some of these references are now added in the revised text as follows:

  1. Kai, D., et al., Engineering highly stretchable lignin-based electrospun nanofibers for potential biomedical applications. 2015. 3(30): p. 6194-6204.
  2. Ma, A., L. Zhou, and J.J.N. Chang, Conversion of lignin-nanofibers to CNFs. 2015. 10(06): p.
    1550092.
  3. Zheng, P., et al., The thermolysis behaviours of the first generation dendritic polyamidoamine. 2009.
  4. Kanani-Jazi, M.H., S. Akbari, and M.H.J.A.P.T. Kish, Efficient removal of Cr (VI) from aqueous solution by halloysite/poly (amidoamine) dendritic nano-hybrid materials: kinetic, isotherm and thermodynamic studies. 2020. 31(9): p. 4018-4030.

Deference in residue weight after thermal decomposition does not provide information about thermal stability. The onset decomposition temperature is used for this purpose.

> We appreciate your comment very much. The onset decomposition temperature and the residue weight at Td for all samples are now added in Table 4.  The related paragraph has also been modified in the text to refer to Td.

How CO2 was produced under inert atmosphere? The mechanical results (E about 80 GPa) are extremely high for polymer materials.

>Thank you for your comment. Some explanations along with references are now added in the revised text for thermal decomposition. CO2 was produced due to the detachment of C-O-C and C=O bonds [62].

> Thank you so much for catching this mistake on the unit of Elastic modulus (E). The correct unit should be MPa, and it has been now applied in the revised version.

Please provide mechanical curves for proving.

>Thank you for your valuable comment. Per your suggestion, Figure 4 (stress-strain curves of lignin and lignin/PAMAM electrospun mates before and after heat-setting procedure) has been added in the manuscript.

Why concentrations 1 and 2% were chosen for preparation of composites.

>Thank you for your question. Regularly, 1% PEO have been added to enhance the spin-ability of lignin-based electrospun fibers. Adding more PAMAM was an original aim of this paper to potentially further improve physical, chemical and mechanical characteristic of lignin/PAMAM. The results revealed, however, that 2% is not possible to spin (due to ensuing excessive solution viscosity; see also resulting material non-fibrous mesh in Fig 1c). 

Minor points

The acceleration voltage in SEM experiment was only 5kV, like it is seen in Figures. However, the range 5-20 kV is given. Fig. 5. Y-axis without units. Please check manuscript for typos.

> Thank you very much for pointing this out. The entire manuscript was double checked and made sure that only 5kV is reported.

> For comparison of the FTIR intensity values, no units are used (Fig. 5. Y-axis is correct).

Round 2

Reviewer 1 Report

The revisions are satisfactory, and they improve the paublication.

No further comments.

Author Response

Thanks for your nice words.

Reviewer 2 Report

The onset decomposition temperature and the residue weight at Td for all samples are now added in Table 4.”

The onset temperature is temperature for beginning of decomposition, which is usually found as temperature for 5% of weight loss. These values should be added to Table 4.

Figure 4 (stress-strain curves of lignin and lignin/PAMAM electrospun mates before and after heat-setting procedure) has been added in the manuscript.”

The curves are not appropriate: 1. They should from 0% of strain. The point for strain at break is not connected with table 4. All curves stop at 4.5%. What is the reason for decreasing of stress at increasing of strain, which can be observed for all samples? It seems that specimens were poorly fixed in the clamps. In this case the results of mechanical measurements are not reliable.

Minor points

“For comparison of the FTIR intensity values, no units are used (Fig. 5. Y-axis is correct).”

This is true. However, X-values (wavenumbers) should be provided in any case.

The precision of temperature like 0.01 K is not believable. (Table 4)

Indicating of excess of digits in precision (like 7.02±1.96) is not the common scientific style. It should be 7±2. Please, correct this through the manuscript.

Author Response

The onset temperature is temperature for beginning of decomposition, which is usually found as temperature for 5% of weight loss. These values should be added to Table 4.

  • Special thanks for your useful comments throughout the review process; the onset temperature for 5% of weight loss has been now added in Table 4.

“Figure 4 (stress-strain curves of lignin and lignin/PAMAM electrospun mates before and after heat-setting procedure) has been added in the manuscript.”

The curves are not appropriate: 1. They should from 0% of strain. The point for strain at break is not connected with table 4. All curves stop at 4.5%. What is the reason for decreasing of stress at increasing of strain, which can be observed for all samples? It seems that specimens were poorly fixed in the clamps. In this case the results of mechanical measurements are not reliable.

  • We are thankful of your valuable tips. There was a minor slippage in the beginning of deformation tests, hence slightly not starting from zero force; we have now corrected the curves in the revised text, along with appropriate caption. Also please note that the curves in Fig 4 are sample results from each mat category (through performed repeats), whereas Table 2 reports average and standard deviation over all performed tests along with their repeats. This has also been clarified in the caption of Fig 4.

Minor points

“For comparison of the FTIR intensity values, no units are used (Fig. 6. Y-axis is correct).”

This is true. However, X-values (wavenumbers) should be provided in any case.

  • Thanks again for your comment. Fig. 6 has been redrawn to include the wavenumber values.

 The precision of temperature like 0.01 K is not believable. (Table 4)

  • Table 4 has been revised according to your comment.

Indicating of excess of digits in precision (like 7.02±1.96) is not the common scientific style. It should be 7±2. Please, correct this through the manuscript.

  • Thanks for your comment. The mechanical characteristics and weight residue values are indicated with two decimal places. Other numbers throughout the manuscript have been reported without decimal in the latest version.

This manuscript is a resubmission of an earlier submission. The following is a list of the peer review reports and author responses from that submission.

Round 1

Reviewer 1 Report

Recommendation: Major revision

Overall, the research on blending lignin and dendrimer is interesting. However, the authors did not clearly address the significance of the research. What will be the application for this blended material?

The writing of the manuscript needs to be improved.

In Figure 2, the fiber diameter unit (nm) is cut in the distribution chart.

There are no standard deviation data presented in Table 3 and no error bar in Figure 6, which makes less meaningful of the statements.

The comparison of viscosity and fiber diameter after 5 days is confusing. The author claimed that the incorporation of PAMAM affects the viscosity and make the electrospinning impossible at 2wt%. However, what is the purpose of testing viscosity in 5 days?

Reviewer 2 Report

This paper is interesting, well argued and fits well with the scopes of the journal.

By focusing on the realization of lignin based electrospun microfibers, it analysis the influences of a polyamidoamine dendritic polymer to improve the electrospinnability as well as the morphological and physico-chemical characteristics of lignin fibers.

Data are clearly presented and conclusions are consistent with both premises and discussion.

Reviewer 3 Report

The manuscript deals with electrospinning carbon fibers consisting of blends of lignin with (aminated) dendrimers. This is an appropriate subject of potential interest to many readers. Unfortunately the manuscript (and the study) suffers from significant flaws. 

The supposed blend consists of a mixture of kraft lignin with 1% dendrimer; a mixture with 2% PAMAM is inoperable as revealed by SEM. The authors fail to examine the blend composition in the usual way, by solvent casting and thermal analysis. The results suggest that the mixture represents a composite structure rather than a molecular blend, at least at a component ratio of 98:2.

The description of the chemistry involved is wrong while being unnecessary:(native)  lignin is not a "network", irregular or not (line 24). (It consists of 50+% aryl--alkyl ethers and is thermally deformable.) The hydrogenation of methoxy groups, and the methanol and methane production discussion (line 175 ff.) needs some critical references.

The conclusions are supposedly "systematically confirmed" (line 339) by FTIR and viscosity, which is an unrealistic statement. The FTIR spectra of lignin vs. lignin/1%PAMAM (Fig. 5a vs. 5b) are identical. For the change of lignin viscosity vs. time, I recommend reviewing Siochi et al., Macromol. 23 1420 (1990). 

Regarding the conclusion statements, (a) the greatest effect on fiber dimension is the heating protocol (to 150 vs. 250C by "setting" vs. "stabilizing"), (b) the mechanical properties of untreated lignin mats are all greater than (heat-set) blends (what is the role of PAMAM and PEO on heat setting?); and (c) the zeta potential increases by only 10% with added PAMAM and needs a statistical verification. It seems to contradict the claim that the lignin/1% PAMAM  fibers have a 45% greater breaking stress than the lignin fibers. Why is heat-setting reducing breaking stress of lignin fibers by half and much less in the presence of 1% PAMAM? Is PAMAM stabilizing lignin degradation? Fig. 4 (number missing on line 191) seems to indicate that the fibers with PAMAM degrade at slightly lower temperatures than lignin fibers. A greater evaluation of thermal behavior might be called for.

A publishable report would require a greater depth in evaluating variables on all levels, esp. regarding the molecular interactions of lignin and PAMAM (which appear reasonable but unsubstantiated; what produces the difference between 1 and 2% PAMAM?).

In its present form, the manuscript is too mysterious to be publishable. It also is in need of extensive language revision. The field of electrospinning of lignin is not new; a thorough review of the current state of art might help strengthen the proposed involvement of PAMAM. But the overall aim the study is certainly of current interest.